# Recent Advances in the Genetic, Anatomical, and Environmental Regulation of the *C. elegans* Germ Line Progenitor Zone

**DOI:** 10.3390/jdb8030014

**Published:** 2020-07-22

**Authors:** Kacy Gordon

**Affiliations:** Department of Biology, The University of North Carolina at Chapel Hill, Chapel Hill, NC 27599, USA; kacy.gordon@unc.edu; Tel.: +1-919-962-2164

**Keywords:** *C. elegans*, germ line, stem cell, progenitor cell, Notch

## Abstract

The *C. elegans* germ line and its gonadal support cells are well studied from a developmental genetics standpoint and have revealed many foundational principles of stem cell niche biology. Among these are the observations that a niche-like cell supports a self-renewing stem cell population with multipotential, differentiating daughter cells. While genetic features that distinguish stem-like cells from their differentiating progeny have been defined, the mechanisms that structure these populations in the germ line have yet to be explained. The spatial restriction of Notch activation has emerged as an important genetic principle acting in the distal germ line. Synthesizing recent findings, I present a model in which the germ stem cell population of the *C. elegans* adult hermaphrodite can be recognized as two distinct anatomical and genetic populations. This review describes the recent progress that has been made in characterizing the undifferentiated germ cells and gonad anatomy, and presents open questions in the field and new directions for research to pursue.

## 1. Introduction

The *C. elegans* distal tip cell was one of the first cells described with stem cell niche-like properties [1], and since its discovery, many features that are shared among diverse stem cell niches have been modeled in *C. elegans.* These have been reviewed comprehensively and recently [2,3,4,5,6], and in the frequently updated community resources WormAtlas [7] and Wormbook [2]. The field has made tremendous progress genetically defining the populations of germ cell progenitors in the distal gonad, with a distal most stem-like population and more proximal mitotically dividing cells that are further along on the path to differentiation. While the Notch signal that maintains the stem-like state of the distal most cells is known, the mechanisms that segregate self-renewing stem-like cells in the distal germ line from their differentiating daughters are not understood. No distinct genetic signatures have been identified that distinguish these populations; rather, they seem to exist on a continuum graded between stem-like and differentiated. Potential explanations for the maintenance of these populations range from their genetic regulation [5] and cell cycle constraints [8] to various structural aspects of the gonad [9,10]. Understanding the control of self-renewal and differentiation is important for several reasons. The *C. elegans* gonad is a model stem cell niche system, and its usefulness as a model depends on how analogous elements of the niche function. The potential to discover new genetic or structural regulators of stem cell differentiation and renewal in this system is high, and these findings may lead to discoveries that are applicable to other stem cell niche systems. The interactions between the genetic systems described in the *C. elegans* germ line and environmental inputs can reveal how the germ line responds to challenges to transit generations. 

## 2. Results

### 2.1. Anatomy

#### 2.1.1. Hermaphrodite Gonad Structure

The hermaphrodite gonad is composed of symmetrical anterior and posterior U-shaped gonad arms, each with a distal tip cell (DTC) at its distal extreme [11] (Figure 1A). The DTC acts as the germ line stem cell niche, and each DTC caps a pool of germ stem cells (GSCs), which is part of the larger germ line syncytium, that comprises about a thousand germ cell bodies arrayed around a central core of cytoplasm, called the rachis (Figure 1B). The rachis and the cytoplasmic bridges that connect it to the germ cells are stabilized by the actin cytoskeleton and associated proteins like anillins [12,13] and others [14,15]. The GSCs are found at the distal end of the gonad, with cells showing genetic and cytological markers of differentiation further proximal (Figure 1A); differentiation for germ cells means entry into the meiotic cell cycle. The undifferentiated germ cells at the distal end of the gonad have been referred to as the “mitotic zone” (contrasting with the differentiated meiotic germ cells further proximal), the “proliferative/proliferation zone” (where active cell divisions are occurring), and the “progenitor zone” that contains both stem cells and non-stem progenitors that have lost their stemness but also have not yet differentiated, as well as cells in meiotic S-phase [6]. I will use this latter terminology. The progenitor zone (PZ) of an adult hermaphrodite comprises ~200–250 of the distal most germ cells in each gonad arm. Proximal to them are the cells of the “transition zone” in which distinct crescent-shaped chromosomes of cells in early meiotic prophase are observed. 

Proximal to its DTC, each germ line is covered by five pairs of very thin somatic gonadal sheath cells (Figure 1A). The distal-most gonadal sheath cell pair, Sh1, covers germ cells from the edges of the DTC to the bend of the gonad (Figure 1A–E) [10]. An arrangement with a somatic gonad sheath cell close to the DTC is also found in the female of the filarial parasitic nematode *Brugia malayi* [16,17], a distant relative of *C. elegans*. This discovery changes the previous model in which germ cells proximal to the DTC and distal to the transition zone were thought to be in a “bare region” where they contacted no somatic cells, only the gonadal basement membrane and occasional thin DTC and sheath projections [11]. Below, this review will further explore how the discovery of Sh1 making extensive contacts with the PZ calls for the formulation of a new model of GSC compartmentalization. 

The Sh2 cells reside at the bend of the gonad. Sh3-Sh5 cells cover the proximal gonad where oocytes form and wait to be ovulated, and make up the contractile apparatus responsible for ovulation [18] (Figure 1A). Sheath cells require CACN-1/Cactin for proper splicing and development, without which worms are sterile [19]. Proximal to the sheath are the spermathecae, which store sperm and through which ovulation pushes the oocytes to become fertilized. After passing through a spermatheca and into the uterus, an embryo will undergo five rounds of cell division and then be laid at the 32-cell stage through the vulva. The entire gonad, from uterus to DTCs, is encased by the gonadal basement membrane [20,21]. 

#### 2.1.2. Gonad Development 

Gonad development is a postembryonic process [22]. The L1 larval gonad primordium is made of four cells, Z1-Z4, which will give rise to the somatic gonad (Z1, Z4) and primordial germ cells (Z2, Z3). Divisions of these cells begin in the L1 larva. The DTCs are born from the second division of Z1 and Z4 and are required for the continued proliferation of descendants of the primordial germ cells. The sheath cells are born from divisions in L3 larva, with the Sh1 cells born approximately ten hours before the other four pairs [22]. The DTCs lead the migration of each gonad arm, first away from the ventral midbody, then with a first turn towards the dorsal body wall, followed by a second turn onto the dorsal body wall, and back towards the midbody. The germ cells proliferate behind the migrating DTCs. Migration ceases at the end of the L4 larval stage. The process of DTC migration has been extensively genetically dissected, and this work has been reviewed elsewhere recently [23,24,25]. 

#### 2.1.3. Distal Tip Cell Structure 

The DTC undergoes a rapid and dramatic remodeling at the larval/adult transition [10,26,27], during which it transforms from a gumdrop-shaped cap in L4 larva to a large, jellyfish-like plexus in adults with elaborated branches traveling proximal among the germ cells (Figure 1B, germ cells not shown). In males, there are two DTCs and they are smaller and not elaborated, though they regulate stem cells in a similar manner [28]. The mechanism by which the DTC elaborates has only just begun to be understood, but the processes grow among, and rely on, the normally proliferating germ cells [27,29], suggesting that it may be related to germ cell proliferation. 

#### 2.1.4. Open Questions about Somatic Gonad Structure and Function

An open question is how the elaborate structure of the distal tip cell relates to its niche function—direct contact between the DTC and germ cells does not seem to be absolutely required for them to remain in a stem-like state [28,31], and long DTC branches reach beyond the stem cell territory [26,27]. The DTC is striking for its complexity and variability—no other cells in *C. elegans* except for the closely associated Sh1 cells take a unique shape in every individual, with even highly elaborated neurons making stereotypical dendritic branches [32]. It is a large cell, with its longest branches routinely reaching 100 microns or more [10]. How it grows from the small cap in larvae to the large, branched structure in the adult [26,27] is not known. The process must involve the production and trafficking of the membrane. DTC elaboration and function likely involve long distance transport within the cell, the establishment and maintenance of distinct membrane domains, and different modalities of communication with germ cells, possibly in both directions. Fragments are shed from the DTC during migration, and as germ cells leave the niche, though how and when this occurs, and whether these fragments interact with germ cells is unclear. The DTC contacts germ cells and the distal sheath cells during larval DTC migration/gonad elongation and in the adult; the nature of these adhesions is only beginning to be understood [33]. How these somatic cells interact with the folded rachis of the germ line (Figure 1F) is beginning to be explored [30]. The DTC contacts and produces basement membrane proteins; how they migrate while encased in basement membrane is not known [24]. 

Because *C. elegans* is optically clear, imaging fluorescently labeled living animals is possible. CRISPR/Cas9-mediated genome editing works extraordinarily well in *C. elegans* [34,35,36,37,38,39,40], allowing for the endogenous tagging of proteins, as well as the stable transgenic marking of cell types. Fluorescently tagging endogenous proteins is the key to seeing fine structures that may be disrupted by the overexpression of transgenes, though these protein fusions must always be validated for functionality [10]. Such techniques allow the circumvention of artifacts introduced by fixation, and allow for the real-time observation of cellular and developmental events, including stem and progenitor cell division [10,41], active sites of transcription [31], DTC migration [42] and elaboration [29], the dynamic growth of Sh1 [10], and meiotic chromosome movements [43]. 

### 2.2. Genetics

#### 2.2.1. Genetic Regulation of the Progenitor Zone

While there is still much to learn about the distal tip cell, its role as the stem cell niche that maintains germ stem cell self-renewal is well-described. Early work on the *C. elegans* gonad determined that ablation of the DTC [1] and genetic loss of Notch signaling [44] causes the loss of mitotically dividing distal germ cell population and the entry of all germ cells into the meiotic cell cycle. Conversely, constitutive activation of the Notch receptor in a *glp-1(gf)* gain-of-function mutant causes a tumorous germ line full of mitotically dividing cells that never differentiate [45,46]. Through subsequent careful genetic analysis, a network of genetic control over cell fate was illuminated (Figure 2A). DSL ligands in the membrane of the DTC (LAG-2 and APX-1) activate Notch signaling in the distalmost germ cells via the GLP-1 receptor [47,48,49]. The Notch signal leads to transcriptional activation of recently discovered (see below) targets *sygl-1* and *lst-1* that encode proteins that partner with PUF/FBF proteins FBF-1 and FBF-2 and—as recently discovered in [50]—PUF-3 and PUF-11 to regulate the size of the progenitor zone [51,52,53,54,55] by inhibiting meiosis via translational repression of target mRNAs, among which are found representatives of the three pathways regulating meiotic entry [55,56,57,58]. 

Proximal to the region of active Notch signaling, meiotic entry proceeds via three pathways—one that inhibits translation of genes promoting the undifferentiated fate, one that promotes translation of meiotic entry genes, and one that causes proteins promoting the undifferentiated fate to be degraded [57]. The *gld-1* pathway inhibits the undifferentiated fate by the translational repression of mRNAs like *glp-1* via RNA-binding proteins GLD-1 and NOS-3 [59]. The *gld-2* pathway directly promotes meiotic entry by GLD-2 poly(A) polymerase that promotes the translation of meiotic entry mRNAs [60] (including *gld-1*) cooperatively with GLD-3 [58]. Finally, a third pathway characterized by protein-level downregulation of mitosis-promoting genes mediated by the SCF^PROM−1^ E3 ubiquitin ligase complex has recently been identified [61] after such a pathway was predicted years prior [60]. This pathway targets for degradation proteins like CYE-1 that function throughout the PZ [62,63], where they limit the activity of other meiotic entry factors like GLD-1 [63]. Robust bodies of literature exist describing the meiotic cell cycle [64] and the differentiation of sperm and egg cells [65,66], but they will not be reviewed here. 

Despite this thorough characterization, there are several immediate areas that suggest themselves for future research. A widely observed phenomenon in this system is redundancy, making synthetic screens of particular promise going forward [50,52,71,77]. Furthermore, while connections within these networks are well-supported, quantitative experiments have recently revealed a substantial role for GLP-1-independent regulation of network members [68]. Other inputs that influence the size of the PZ are signals from the Sh1 pair of somatic gonad sheath cells. These include an unknown proliferative cue that acts during larval germ line expansion [78] and possibly also (or a different cue) acts during adulthood [10], as well as gap junction signaling [74]. Many steps of this system are under the control of systemic and environmental cues (see Environment, below). 

Finally, the way this program plays out over time is not fully understood. Ample evidence suggests differences in the larval germ line and that of the adult, and between the establishment of the adult niche and its maintenance over time [41,79,80,81,82], so correlating those temporal changes with gene regulatory changes is an essential next step. 

#### 2.2.2. Identification of the Link between Notch Signaling and PUF/FBF

The Notch pathway acts through LAG-1 CSL DNA binding protein as a transcriptional effector [68,83], and activates the target genes *sygl-1* and *lst-1* [31,68,84]. Exhaustive genetic screening for germ line proliferation defects and sterility had failed to turn up the link between Notch signaling and FBF-mediated stem cell fate. Two linking genes—*lst-1* and *sygl-1*—were recently identified by an elegant candidate screen for genes transcriptionally regulated by Notch signaling and post-transcriptionally by FBF binding that function redundantly to maintain GSC self-renewal [52]. They were later validated as quite possibly the only direct targets of Notch signaling that mediate germ stem cell fate by genome-wide approaches, including the direct binding of promoters by LAG-1 and RNA-seq for GLP-1-responsive transcripts [68]. 

These genes encode proteins not found outside of Caenorhabditis nematodes and both have intrinsically disordered regions (IDRs), but no apparent homology to one another [52]. The genes act sufficiently and partially redundantly to promote GSC self-renewal [52,53], however each single mutant has a somewhat different effect on the total number of cells in the progenitor zone [53]. Moreover, *sygl-1* and *lst-1* mRNAs are likely targets of FBF regulation [56,85], while SYGL-1 and LST-1 proteins are components of the FBF regulatory complex [53]. Their mRNAs immunoprecipitate with FBF-1 [56]. These genes do not encode components of Notch signal transduction [52]. LST-1 regulates germ stem cell self-renewal via physical interactions with FBF, and it is likely that SYGL-1 (which also possesses IDRs and a KXXL FBF-binding motif) does as well [54]. They regulate self-renewal by co-regulating FBF target mRNAs in a complex with FBF proteins [53]. One of the target mRNAs that is prevented from being translated by SYGL-1/LST-1/FBF is the mRNA that encodes GLD-1 protein [53], the primary repressor of stem cell self-renewal. These data situate these new regulators of stem cell self-renewal downstream of Notch signaling and upstream of the repression of meiotic fate via the inhibition of *gld-1* translation (where it likely co-regulates other FBF targets like *gld-2* and *gld-3*).

#### 2.2.3. Open Questions of Notch Signaling in the Germ Line

Two direct Notch signaling targets that interact with FBF to prevent meiotic entry have been identified; other questions remain. While canonical Notch signaling is contact-mediated via membrane bound ligands, it is not clear that direct DTC-germ cell contact is required for active Notch signaling in every germ stem cell. Meanwhile nascent *sygl-1* transcripts are observed almost exclusively in germ cells contacting the DTC in adult hermaphrodites [84]; evidence from males demonstrates that germ cells can activate Notch targets at a distance of 1 germ cell diameter or more from the DTC [28]. A new tool that can visualize in vivo transcriptional bursts of *sygl-1* mRNA production by recruiting GFP to its introns using MS2 [31], combined with the first functional tag of the DSL ligand LAG-2 [33], may help shed further light on this subject. LAG-2, a necessary [47,86] ligand that is expressed by the DTC to activate Notch signaling in the germ cells, is punctate and nonuniform across the surface of the DTC, and is localized to some but not all DTC branches [33]; active transcription of direct Notch pathway targets seems primarily (but not absolutely) restricted to those germ cells in contact with the DTC [84]. The spatial pattern in which LAG-2 activates GLP-1 receptors and how long the GLP-1 Notch intracellular domain (NICD) remains in the nucleus—specifically, whether it perdures in cells after they divide—have yet to be described. 

Following on from that question, several secondary targets of Notch signaling in germ cells have been identified, without known roles in the germ line [68]; any activity in the stem cell self-renewal program is yet to be discovered. Perhaps more crucially, a downstream target of Notch activation that transcriptionally regulates other genes that are indirect targets of Notch (like *lip-1* and *fbf-2*) has yet to be described. Another regulatory linkage that has yet to be fully explicated is that between Notch signaling and the gap junctions that form between the somatic gonad (DTC and Sh1 cells, with expression in either cell type sufficient to rescue germ cell proliferation) [74]. These pathways have complicated epistasis, with the innexin mutations epistatic to both loss of function and gain of function mutations in the Notch pathway [74]. Targeted searches for these missing players will complete the picture.

Recent findings also raise several evolutionary questions. Despite the broadly shared mechanism of Notch signaling as the mediator of stem-cell fate at the top of the regulatory hierarchy and the downstream effectors like PUF/FBF that are common factors acting in animal stem cells [87], the links between these two conserved genetic programs are genes that lack sequence homology outside of Caenorhabditis nematodes. Within the clade, lineages have repeatedly evolved germ lines with the ability to produce male and female gametes via the independent recruitment of at least one essential regulator to different roles in gamete sex determination [88,89], demonstrating the evolvability of germ line regulation. Most documented germ line gene evolution in these self-fertile nematodes has been via gene loss [90,91,92], so the acquisition and essential function of relatively new genes is a new direction to explore in this clade with unique reproductive strategies. 

#### 2.2.4. The Germ Stem Cells

The GSCs of *C. elegans,* unlike the other cell lineages of *C. elegans*, are not fixed in their lineage, number, or precise positions. GSCs are nestled in the distalmost plexus of DTC cellular processes [26,27]. They are characterized by nuclear LAG-1 protein [68], and the active transcription of direct targets of Notch signaling *sygl-1* and *lst-1* [31,54,68,84,93]. They appear to be isolated from the rest of the germ line by a diffusion barrier in the cytoplasmic core of the germ line rachis [9], and extend as far proximal as the interface between the DTC and Sh1 [10]. LST-1 is restricted to the distalmost 40–50 germ cells [53], perhaps due to the more restricted region of active *lst-1* transcription [84] and LST-1 autoregulation of its own mRNA [54]; SYGL-1 is present in an overlapping, larger pool of over 100 germ cells [53]. To what extent this difference further subdivides the stem cell zone remains to be seen and will be discussed below. At the edge of the stem cell zone, germ cells reside at the interface between the DTC and Sh1 cells [10] (Figure 1E).

#### 2.2.5. Mitotically Dividing, non-Stem Progenitor Cells

Germ cells appear primarily to leave contact with the DTC by dividing asymmetrically at the DTC-Sh1 interface [10], and cells proximal to this interface are no longer marked by *sygl-1* promoter activity and have an easily detectable GLD-1::GFP signal [10]. These germ cells retain their undifferentiated state due to regulatory cascades downstream of Notch signaling—without Notch activity, cells in this region differentiate [94,95]. They are farther along the path to differentiation than the distal-most cells; when mitotic progression is inhibited by an *emb-30(ts)* allele, these cells differentiate instead of arresting in metaphase of mitosis [95]. They can be recognized by relatively lower levels of Notch targets, high levels of REC-8 [60] and CYE-1 [68], and intermediate levels of GLD-1 [10,30,53,96]. Their position in the gonad falls between ~10–20 gcd from the distal end, and roughly 100–150 cells fall in this region in young adults. 

No unique molecular signature has been detected to mark these cells as distinct from the germ line stem cells, other than relative levels of genes regulating self-renewal and differentiation, as described above. One candidate that should be explored for its relationship with the non-stem progenitors is *lip-1*. While not a direct target of Notch signaling [68], *lip-1* is downstream of Notch signaling, and *lip-1* mRNA is negatively regulated by FBF [67]. LIP-1 protein therefore accumulates outside of the GSCs in proximal proliferating cells, where it is thought to inhibit MAPK signaling-induced progression into the meiotic cell cycle, along with and possibly downstream of OGR-2 [67,97]. The roles of MAPK in the germ line are complex—including a non-essential promotion of mitotic fate at the expense of apoptosis [98] that complicates the understanding the apparent importance of excluding MAPK from the PZ—and are the subjects of active inquiry [77]. However, the regulatory circuit that governs LIP-1 exclusion from the distal zone—activation downstream of Notch signaling *and* mRNA-mediated translational repression by FBF, may turn out to be widespread in the non-stem progenitor population, as it neatly primes germ cells to begin accumulating gene products just outside of the region of highest Notch target expression (Figure 2B). The gene products that subsequently inhibit entry into meiosis may play an important role in setting the size of the PZ.

A recently discovered feature that distinguishes non-stem progenitors from GSCs is their close association with the Sh1 pair of gonadal sheath cells [10]. The Sh1 cells cover most of the PZ (Figure 1A), with roughly one third of PZ germ cell divisions observed to be in contact with Sh1 but not with the DTC [10]. Thus, the PZ is anatomically divided at the DTC-Sh1 interface, with stem-like cells falling distal to that region and non-stem progenitors under the Sh1 alone. Perturbing the distal position of Sh1 cells by inhibiting actin cytoskeletal remodeling shifts the distalmost region of GLD-1::GFP protein accumulation proximally and reduces proximal PZ germ cell divisions, though it does not change the overall size of the PZ [10]. Together, these results indicate that the Sh1 cells support germ cell proliferation in the proximal PZ. This effect seems to be additive with the pro-proliferative influence of the DTC, as germ cells at the DTC-Sh1 boundary divide more frequently than those either under the DTC alone or Sh1 alone [10]. Sh1 cells’ ability to promote germ cell proliferation via an unknown mechanism has been described in larval stages [78,99], and insulin signaling has been excluded as the molecular basis of the signal [69]. The identity of the signal, and whether the same signal functions in the adult, remain to be seen. 

#### 2.2.6. Cells Undergoing Meiotic Entry

Germ cell progenitors entering the meiotic cell cycle are marked by robust levels of GLD-1 protein [8], the absence of Notch pathway-mediated transcription [84], and negligible levels of direct Notch targets LST-1 and SYGL-1 [53] and progenitor zone proteins CYE-1 [68], REC-8 [60], and WAPL-1 [61,81]. While this review will not interrogate the regulation of progression through the meiotic cell cycle, recognizing cells in early stages that have left the mitotic cell cycle is crucial to studying the rest of the progenitor zone, in which some cells are stem cells, and some are irreversibly committed to differentiation but have not yet adopted the meiotic fate [8,95]. 

### 2.3. Topics Arising in Germ Line Regulation 

#### 2.3.1. Spatial Restriction of Cell Fate Determinants

A striking feature that emerges from the study of the genetic control of the PZ (Figure 2A) is the abundance of post-transcriptional and post-translational control, cross-regulation, and autoregulation among these factors. LST-1 negative autoregulation at its mRNA and protein levels [54], GDL-1 repression of LAG-1 [68] and GLP-1 [100], FBF negative autoregulation [51], among other regulatory loops, set limits on the extent of gene product localization. It is possible that these genetic interactions are sufficient to create stable domains on their own. However, structural features of the gonad likely also function to maintain and/or establish these asymmetries. 

One mediator of spatial restriction is the asymmetric position of the DTC at the extreme end of the distal germ line. The two direct transcriptional targets of Notch were found by smFISH and live imaging to be activated in a probability gradient in the distal most germ cells [31,84] in close association with the DTC. Furthermore, there is evidence of *sygl-1* promoter expression only in germ cells within the plexus of the DTC [27], rather than just those located at the distal extreme of the germ line, suggesting that the shape of the DTC niche cell is crucial to stem cell regulation. A CRISPR/Cas9-endogenously tagged Notch ligand LAG-2 on the DTC cell surface [33] suggests that non-uniform localization of the ligand on the DTC may help restrict the expression of GLP-1/Notch targets; this may be a partial answer to the question of why only germ cells in the plexus of the DTC, and not along all of its long branches, have active notch signaling. The movement of germ cells out of the distal region of the gonad is too slow to invoke a positional mechanism by which proximal displacement leads to the loss of Notch signaling [84], however, so the presence of the DTC at the distal end is only a partial explanation for the localization of GSC factors. 

Recently, the Sh1 distal gonadal sheath cells have been discovered to maintain thin, membranous, distal protrusions that abut the densest region of DTC processes [10]. These protrusions appear to grow in coordination with the division of germ cells at the DTC-Sh1 interface, having been observed to extend into the cleavage furrows of dividing germ cells so that one daughter cell remains at the interface and the other associates with the Sh1 alone [10]. These germ cells associated with Sh1 and not the DTC express GLD-1::GFP at higher levels [10], suggesting asymmetric cell fates as well as asymmetric contacts across the DTC-Sh1 interface. Thus, it may be that the presence of Sh1, in addition to the absence of the DTC, is the determining factor of stem cell niche exit. 

Even with genetic circuitry and external structures that facilitate the subdivision of the PZ, the existence of the germ line rachis—a shared core of cytoplasm—complicates the notion of signals being neatly restricted. The rachis could be a conduit for molecular specifiers of cell fate (mRNA, protein, small molecules). Some studies have found that neighbor-to-neighbor transfer of cellular contents through the rachis is minimal [84], especially during M phase, when cells close their cytoplasmic bridges to the rachis [30]. However, there is evidence that concentration gradients of factors across the cell bodies and the rachis that they border are the same [30], implying diffusion in at least one direction. Local signaling among germ cells is also one proposed explanation for the observed clustering of germ cell divisions in space and time [101]. Aside from the rachis, germ cells show widespread cell-cell adhesions to one another via cadherin/catenin and L1CAM [33], though the significance of these adhesions is not known. Local signaling among near neighbors along the rachis of the germ line is an open question. 

Diffusion barriers in the rachis prevent the flow of molecules from one region of the germ line to another [9]. These barriers may be synonymous with folds of the rachis itself that result from abundant proliferation in the PZ [30]. Diffusion barriers appear to be formed by constrictions and folds of the rachis. The limit on diffusion they allow from the proximal side correlates spatially with the distal extent of a GLD-1::GFP marker of differentiation, and observations and modeling propose a role in limiting the extent of Notch signaling, suggesting that diffusion barriers could help restrict cell fates within the PZ [9]. Such barriers could also act as a brake on proliferation: proliferation in excess of exit from the PZ leads to germ line folding [102], which may prevent the diffusion of pro-proliferative cues, allowing proximal PZ cells to enter meiosis. The relationship between rachis folds/diffusion barriers and Sh1-germ cell interactions have yet to be explored. 

#### 2.3.2. A Role for Gonad Structure in Patterning the Germ Line

The *C. elegans* germ line is considered to be a model for a population-based mode of stem cell maintenance, predicated upon GSCs making exclusively symmetrical divisions [6]. The discovery of asymmetric germ cell divisions across the DTC-Sh1 boundary [10] raises the possibility that, along with symmetrically dividing GSCs under the DTC, there is a separate group of cells that depend upon the asymmetric arrangement of their daughter cells between the DTC and Sh1 to inform the self-renewal vs. differentiation decision (Figure 1E). Additional evidence suggests that these cells at the DTC-Sh1 boundary may be distinct genetically as well as anatomically.

I propose a model (Table 1) in which non-meiotic PZ cells are made up of three anatomically and genetically distinct groups, rather than a single population that is graded from stem-like to differentiated with a continuum of states in between. A major benefit of this model is that it incorporates the anatomical variation that may influence germ cell fates beyond row position alone, so the “row” ranges that follow should be understood to reflect not only uncertainty due to comparing across different experiments but also true variability in *C. elegans* rachis folds, DTC, and Sh1 cell positions.

The distalmost ~2–4 rows of GSCs are completely enwrapped by the DTC plexus [26,27], lack contact with Sh1 [10], cycle more slowly [101,103], have lower DNA content [82], have the lowest GLD-1 levels [8,30], have *lst-1* and *sygl-1* active transcription sites and gene products [84], divide symmetrically [10], and retain their distal position over time [104]. The next-most-distal cells fall on the DTC-Sh1 interface in rows ~5–9 [10], divide more frequently [10,101,103], have higher DNA content [82], have faint but above base level GLD-1 protein signal [8,10,30], lack *lst-1* transcription sites but have *sygl-1* transcription sites and both proteins [84], and can divide asymmetrically [10], with descendants that leave the niche [104]. Finally, the cells in the proximal PZ (rows ~10–15) lack contact with the DTC (except some long projections in older worms), are completely covered by Sh1 [10], divide more slowly [101,103] (but note: they still initiate cell divisions, see below), have the highest DNA content [82], show the strongest GLD-1 protein signal outside of meiotic cells [8,10,30], lack active Notch target transcription [31,84], but may remain active for downstream Notch targets due to protein perdurance [10,53], may be marked by proteins like LIP-1 [67] under the logic of the circuit shown in (Figure 2B), and divide symmetrically [10]. None of these territories have set positions either measured in rows or microns from the distal end; rather, they likely depend on their position along the rachis [30] and somatic cell contacts [10,27] and will vary among individuals and even among neighbors at a given proximodistal position in the germ line (Figure 1E). Further regionalizing the stem cell pool may contribute to fine-tuning proliferation rates among progenitor cells to minimize mutation, as stem cell cycling is related to mutational acquisition [79,80].

#### 2.3.3. Testing the Model with Existing Data

The existence of a transit amplifying (TA) population in the germ line has been contentious. TA cells are stem cell progeny that have left the niche but have not yet differentiated and instead continue to undergo some number of mitotic divisions before differentiating [106,107]. Experiments using *emb-30* mutants blocked for the metaphase-to-anaphase transition at the restrictive temperature show that a distal 6–8 rows of germ cells arrest in metaphase and fail to acquire markers of differentiation as rapidly as more proximal germ cells in the PZ [95]; the more proximal cells also rapidly enter meiosis upon refeeding after starvation [105]. Likewise, the removal of Notch signaling via temperature sensitive *glp-1(ts)* shows a proximal-to-distal wave of differentiation, however, the same experiment repeated with different markers of differentiation contested the existence of a wave [94]. Instead, that study suggests that divisions in the proximal PZ results from a mitotic program that cannot be diverted once mitotic S phase initiates, leading to divisions completing at some distance from the DTC rather than the initiation of cell division outside the niche [94]. 

The new model presented here resolves some of these disagreements. In *emb-30* mutants [95], distal and interface germ cells can still actively receive Notch ligand from the DTC. According to the territories described in [84], this may correspond to a pool of cells with active Notch target transcription, but not the broader pool of cells that are positive for Notch target transcripts and proteins [53]. Those non-stem progenitors differentiate in place because they lack continuous active Notch signaling. Under wild-type conditions, non-stem progenitors would be analogous to TA cells that divide around once after leaving the niche. This conclusion is based on the fact that the PZ cells that contact Sh1 and not the DTC appear to initiate that contact at birth [10], not in the middle of a mitotic cycle as suggested by [94], and it is compatible with the finding that proximal PZ cell populations increase in size 1.5-fold in 8 h [104].

When the gene encoding the Notch receptor is downregulated in a *glp-1*(*ts)* mutant, one would expect that the GSCs are primarily affected by the mutation—the more proximal PZ cells are not in fact experiencing a different regulatory regime in the temperature-shifted mutant—they were negative for active Notch signaling to begin with and likely retain their Notch downstream targets after the temperature shift. These cells go on to differentiate in place without proximal cell displacement that would normally be caused by proliferation of more distal cells (due to falling mitotic index in *glp-1(ts)* mutants [94])—the wave of differentiation that in wild-type is temporospatial in the mutants becomes strictly temporal. Estimates of the rate of movement at one germ cell diameter per hour [103,104] means these cells are several rows further distal than would be expected under normal conditions. The data in Figure 4 of [94] show a proximal-to-distal sweep of differentiation as distal cells that lost *glp-1* transition from an active Notch to a residual Notch to a no-Notch state over time. Notch activation is dynamic and stochastic, and some GSCs actively express *lst-1* while others do not [31,84]; therefore, the GSCs may differ in the amount of residual Notch signal transduction components or Notch targets that perdure after the receptor is downregulated by temperature shift, which may cause variation in their otherwise closely timed meiotic entry. The remaining disagreement of these studies seems to be the overall size of the stem-like pool (five rows vs. ten), and its location in strict rows vs. more irregular territories. The first point is addressed by a new model that differentiates between the zone in which *lst-1* and *sygl-1* are actively transcribed, and the zone in which *sygl-1* alone is actively transcribed (the genes may have different requirements for Notch levels and therefore different sensitivities to the loss of *glp-1*), and the second point is addressed by our better understanding that “rows” of germ cells are not the most anatomically meaningful way to describe their positions [10,30].

More evidence supporting distinctions among PZ cells comes from studies of assaults on the germ line like reproductive aging and starvation. In the case of aging, all three PZ pools shrink over time, likely reflecting the dependence of all PZ cells on Notch signaling to prevent differentiation [81]. This dependence on Notch signaling is broken during starvation; in starved animals, germ cells in all compartments become quiescent for cycling and do not progress through meiosis [105], however, upon refeeding, some proximal PZ cells rapidly enter meiosis while the more distal cells do not—is the variation among proximal PZ cells explained by some being non-stem progenitors and others being interface GSCs? Short of starvation, dietary restriction also slows the mitosis of PZ cells due to its effect on the spindle assembly checkpoint [41], though which PZ cells are most affected is not known. 

Next steps suggested by this model would be investigating differences in transcriptional regulation for *lst-1* and *sygl-1*. The analysis of both grossly normal *sygl-1* and *lst-1* loss-of-function single mutants [52] and the expanded PZs of gain-of-function mutants [53] for characteristics like division rate and DNA content would reveal whether the aspects of stem cell cycle regulation that differ between the distal-most and interface GSCs in wild-type worms correlate with their *lst-1*-positive or -negative transcriptional state. Along those lines, distinguishing among LST-1/FBF and SYGL-1/FBF targets might further delineate the stem cell regions. Testing other environmental or life history conditions that elicit different responses from different groups of PZ cells, and investigating the role of Sh1 in establishing or maintaining these territories, could also be revealing. 

#### 2.3.4. Environmental Inputs into the Gonad and Germ Line

The size of the progenitor zone and gamete production are highly responsive to environmental and systemic cues. Such cues recently reviewed in [6] include nutrition and starvation, chemical signals from other worms, and aging. The pathways that convey these signals to the germ line include insulin signaling, TOR signaling, and TGF-β signaling [6,108]. Some of these signals act through the nervous system; while nervous system function in environmental sensing is well documented, its mechanism of action on the germ line is less so. 

One of these signals is the male ascaroside pheromone ascr#10, which delays the age-related loss of hermaphrodite PZ cells and accelerates somatic aging [109]. The germ line effect requires the ASI neuron in the head, and likely requires the expression of DAF-7 TGF-β ligand therein, as *daf-7* mutants lack the response [76]. DAF-7 in ASI also signals under replete conditions through TGF-βR DAF-1 expressed on the DTC [110]; in the DTC, active TGF-β signaling relieves DAF-3/DAF-5 repression of *lag-2* and upregulates its transcription in L4 animals [111]. The apparent coalescence of these two sets of environmental cues on ASI/DAF-7 raises the question of whether that neuron is unique in communication with the germ line or its niche. However, the effect of ascr#10 has also been shown to require serotonin, specifically serotonin produced in response to the pheromone by HSN and NSM neurons via the expression of *tph-1* [75]. This signal requires the serotonin receptor gene *mod-1*, and *mod-1* mutants can be rescued for their ability to respond to ascr#10 by restoring *mod-1* function to neurons AIY or RIF, but *not* to ASI [75]. Remarkably, the potentiation of adult hermaphrodites to respond to ascr#10 requires egg laying activity via a mechanism that reports reproductive status through the vm2 muscles to the nervous system [112]. Therefore, not only the ASI neuron, but other neuronal circuits directly related to reproductive behaviors, modulate PZ size. They do this not only by detecting external chemical signals via well-studied sensory processes, but through an internal surveillance of behavior via a reafference-like mechanism. Signaling from within the gonad to the soma is also responsible for the production of pheromone cues. Oocytes signal to the soma to induce the production of a hermaphrodite pheromone that attracts males in the absence of self-sperm [113]. The relationships among pheromone signaling, the germ line and somatic gonad, and the nervous system are complex. The ability to integrate environmental and systemic cues to control reproduction is crucial, especially as these cues may co-occur in nature—for example, the combination of recovery from heat stress and the presence of potential mates [114]. How animals integrate the great variety of cues they are exposed to—which cells integrate the signals and whether they converge on single points of integration, how conflicting cues are resolved, and which take precedence over which others—all remain to be seen. 

#### 2.3.5. The Somatic Gonad as a Relay Point

The close association between the progenitor zone and the gonadal sheath offers a pathway of direct cell-cell signaling to relay information from the proximal gonad—a tissue which communicates directly with the outside environment (via sperm) and the nervous system—to the progenitor zone. The Sh1 cells contact the PZ cells, including some GSCs, as well as the DTC niche cell, throughout development [10]. On their proximal side, they have gap junctions to Sh2 cells, which are likewise connected to Sh3, Sh4, Sh5, the spermatheca, and the uterus in sequence [11,74]. This is the only cell-cell junctional pathway, aside from the germ line itself, connecting the PZ to other parts of the animal. Results demonstrating that egg-laying itself serves to alter the responsiveness of the germ line to certain signals [112] raise the question of which cues from the proximal gonad—the abundance of sperm, the packing of oocytes, the peristaltic waves that ovulate eggs through the spermatheca—can feed back to the PZ, perhaps through a sheath relay to distal Sh1 processes. While these proximal gonad tissues are not thought to be directly innervated (the vm2 muscles are, and are required for the effect of the egg-laying response on the PZ [112]), it may be possible that gap junctions among somatic gonad cells could serve to electrochemically couple them in the absence of neuronal input. 

Whether endocrine signals can diffuse through the gonadal basement membrane (BM) is an open question—there is evidence that such signaling occurs via DAF-7 [111], and TGF-β is among the growth factors that have been reported to be tethered to BM by BM associated proteins via heparan sulfate glycosaminoglycan chains [115]. Downstream of TGF-β signaling, the Hedgehog-related signaling factor WRT-10 produced by the hypodermis may directly bind Patched receptors on the oocytes and germ line [108]. However, RNAi that causes breaks in gonadal BM reveals that the wild-type BM serves to limit signaling at a distance, as in the absence of BM, muscle cells actively envelop the exposed germ cells [33]; the BM surely restricts some diffusible cues that could act between muscle and germ cells. Which signals reach the germ cells directly, and which are transduced through the somatic gonad, and under what conditions and with what effects, are not fully understood.

Outside of the newly discovered far distal region of Sh1-germ cell interaction, the sheath is primarily known to interact with differentiating germ cells [73,102]. In fact, contacts between the proximal sheath and germ cells that have been genetically delayed in meiotic entry develop proximal germ line tumors [116]. Because Sh1 is now known to be in contact with mitotic germ cells in the wild-type [10], this hints at important genetic distinctions between Sh1 and the other four pairs of sheath cells—genes with unique expression in Sh1 could perhaps be identified by RNA-seq or enhancer-trap experiments. Furthermore, regionalization surely exists within adult Sh1 cells, as their distal ends appear to promote proliferation while their proximal ends interact with meiotic cells undergoing apoptosis [73,102]. All of the open questions that applied to the DTC structure apply to Sh1 as well. How does such a large, variable, elaborate cell make so much membrane, grow continuously at its distal end, and regionalize the localization of different factors, and how do these processes contribute to its regulation of the germ line? Very few molecular genetic systems are as well understood as the *C. elegans* germ line fate specification pathway, and *C. elegans* has an anatomy and a toolkit of techniques that perfectly suit it for live imaging. The next wave of discoveries about the fine cellular and molecular regulation of stem cell fate will integrate and build off of previous innovations.

## Figures and Tables

**Figure 1 jdb-08-00014-f001:**
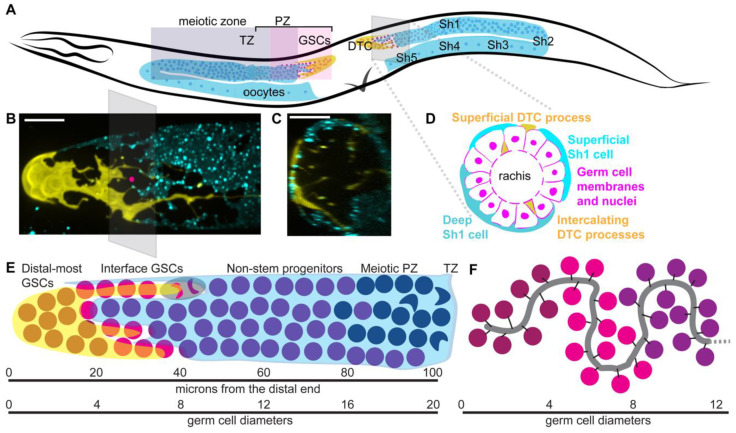
Structure and development of adult hermaphrodite gonad (**A**) Schematic of adult hermaphrodite gonad with distal tip cells (yellow, DTC), sheath cell pairs 1–5 (only superficial cell of each pair shown, cyan); somatic gonad cells are labeled in posterior gonad arm. In the anterior gonad arm, germ cell regionalization is shown in shaded boxes with the distal progenitor zone (PZ) containing germ stem cells (GSCs, pale pink), non-stem progenitor cells (bright pink) and cells in S phase of meiosis I (purple). The transition zone (TZ) is where crescent shaped nuclei of meiotic leptotene/zygotene stage are visible; this is a common visual landmark that can be detected with DAPI staining and does not rely on genetic markers. All meiotic germ cells shaded purple. Oocytes shown in proximal gonad. All germ cell nuclei are magenta. Head to left, dorsal up, vulva (V) at bottom. Gray pane shows angle of cross section shown in (**B**). (**B**) Confocal Z-projection through distal gonad with DTC in yellow (single copy transgene *lag-2p::mNeonGreen::PLC^dPH^*) and Sh1 in cyan (*mKate::inx-8* endogenously tagged protein). Gray pane shows angle of cross section shown in (**C**,**D**), with pink dot at center of stack. (**C**) Cross section of (**B**), Z-projection through 10 microns of the distal gonad around pink dot shown in (**B**). (**D**) Schematic of cross section through PZ with germ cells (magenta), DTC processes (yellow) and Sh1 cells (cyan). Note both superficial and deep Sh1 cells pictured in cross section. Scale bars 10 microns. Pictures by K. Gordon, adapted from [10]. (**E**) Sub-regionalization of the germ line. The DTC (yellow) and Sh1 (cyan) compartmentalize the non-meiotic PZ cells. Under the DTC alone (maroon) are germ stem cells (GSCs) that divide symmetrically; at the DTC-Sh1 interface, additional GSCs (magenta) divide asymmetrically; see example cartoon of cell shaded in gray with anaphase chromatin condensations colored by future daughter cell fates; under Sh1 alone are non-stem progenitor cells (purple) may undergo an additional division outside the niche before entering the meiotic cell cycle (navy). Approximate positions relative to distal end shown, with germ cell “rows” depicted roughly to scale, and to scale with micrograph in (**B**). (**F**) Example of convoluted path of rachis after [30], also shown to approximate scale. The interaction between the rachis folds and the Sh cells is not known; the intercalating DTC processes (not shown) follow the path of the rachis [30].

**Figure 2 jdb-08-00014-f002:**
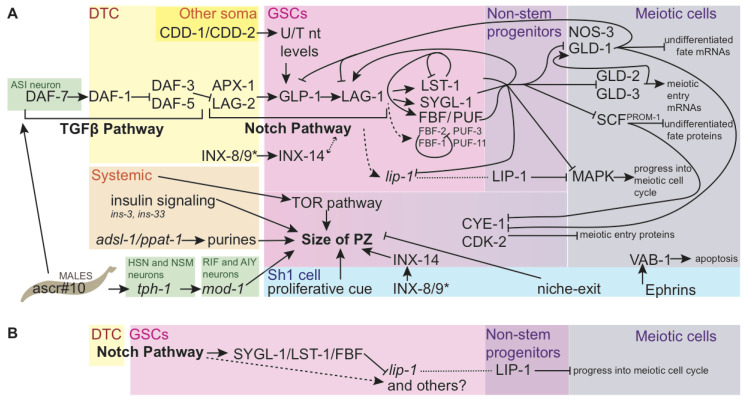
Genetic regulation of cell fate and size of the progenitor zone. (**A**) Genetic regulation of cell fate and size of the progenitor zone. Direct connections are indicated with solid black arrows. Indirect regulation is shown by dashed arrows. Pathway members are clustered with those that are thought to interact at the same step in the pathway, but lines originate and terminate only where there is genetic evidence of interaction. Coalescing arrows show cooperative activity at the next step of the pathway. GLP-1 Notch intracellular domain (NICD) is not shown separately. Colored boxes show sites of action for the genes and proteins named within. Inputs are also shown that affect the size of the PZ, but for which pathways of action have not yet been fully explained; the size of the PZ may be affected by germ cell progress through differentiation as well as proliferation. While regulation is occurring at multiple steps of transcription, translation, and protein-protein interaction, the illustration names proteins in most cases; *lip-1* is an indirect target of Notch in the stem cell region, but is post-transcriptionally repressed by FBF complex there, leading to LIP-1 accumulation in proximal undifferentiated germ cells (see (**B**), [67]). The many genes that regulate the cell cycle in mitotic germ cells are not shown here, nor are components of the meiotic machinery. The many known components of the insulin and TOR pathways are also not shown. FBF targets from [56]. LST-1 autoregulation from [54]. SYGL-1/LST-1 promotion of FBF-2 and LAG-1, and GLD-1 repression of LAG-1 from [68]. Insulin ligands from [69]. SCF^PROM−1^ pathway from [61]. CYE-1 repression of GLD-1 from [62,63], repression by GLD-1 from [62,70]. Nucleotide biosynthesis from [71,72]. Ephrin-apoptosis pathway from [73]. Innexins from [74]; * indicates that innexin presence is required in at least one of DTC or Sh1 (innexins’ role in potentiating germ cells for differentiation is not shown). PUF-3 and PUF-11 from [50]. Direct coregulation of *gld-1* mRNA by SYGL-1 and LST-1 with FBF from [53]; interaction between the direct targets of Notch signaling are shown in cooperation with FBF for other known FBF targets because SYGL-1 and LST-1 are redundantly required for Notch signaling to maintain the stem cell pool [52], though the roles of SYGL-1 and LST_1 in these regulatory events have not yet been experimentally validated. Male signals from [75,76]. (**B**) Simplified sub-circuit showing how indirect transcriptional targets of Notch signaling that are post-transcriptionally inhibited by SYGL-1/LST-1/FBF complex can restrict protein products to the non-stem progenitor zone.

**Table 1 jdb-08-00014-t001:** Anatomical, genetic, and cell-cycle features of non-meiotic PZ cells.

Attribute	Distal-Most GSCs	Interface GSCs	Non-Stem Progenitors	Source
Average position	Distal 2–4 rows, ~0–20 μm	Rows 5–9, ~20–45 μm	Rows 10–16, ~45–100 μm	[6,10]
Contact with DTC	Enwrapped by plexus	Major branches	Longest branches only	[26,27]
Contact with Sh1	None	Contact with distal edges	Uniformly covered	[10]
Division rate	Lowest	Highest	Lower	[101,103]
Division type	Symmetric	Asymmetric	Symmetric	[10]
DNA content	Lowest	Higher	Highest	[82]
GLD-1 abundance	Lowest	Faint	Highest of non-meiotic cells	[8,30]
*lst-1* expression	Actively transcribed	Some transcript/protein	Less transcript/protein	[31,84]
*sygl-1* expression	Actively transcribed	Actively transcribed	Less transcript/protein	[31,84]
*emb-30(ts)* result	Blocks at metaphase	Blocks at metaphase	Differentiates in place	[95]
*glp-1(ts)* result	Differentiates last	Differentiates second	Differentiates first in wave	[94,95]
Response to environment	Shrinks during aging; *glp-1* independent during and resumes mitotic cycle after starvation	Shrinks during aging; *glp-1* independent during and resumes mitotic cycle after starvation	Shrinks during aging; *glp-1* independent during and differentiates after starvation	[81,105]
Additional notes	Retains position 8+ hours	Descendants leave niche	Accumulates LIP-1 protein	[67,104]

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
