# Peer review of "Recent Advances in the Genetic, Anatomical, and Environmental Regulation of the C. elegans Germ Line Progenitor Zone"

_jdb, 2020, doi:10.3390/jdb8030014_

Round 1
Reviewer 1 Report
This is a well-written and incisive review of the C. elegans germline. The description of the somatic gonad is well done, pointing out recent work on the sheath cells that changes some of how we think about the somatic cells associated with the gonads. The review synthesizes a lot of complicated genetic data and makes sense of it. I have only a few minor comments that are intended to make the work accessible to a wider audience.
In Figure 1 panel C is not particularly informative, I understand the idea, but at first I thought it might be the male, then realized that it was only one of the gonads. Given the rich information about the structure and the exceptional image of the DTC/Sheath, panel C stands out. I would argue that the text above does a sufficient job of describing the migration, and that panel C is unnecessary. Alternatively, simply having arrows inside the worm that describe the path might be easier to interpret than what seems to be a pictorial time-lapse
I think an additional image describing the domains of the germline might be helpful, to be able to visualize the progenitor, transition and meiotic regions.
Please define GLP NICD (Line 165) when it first appears.
In lines 216-217, why does the quadruple mutant forming tumorous germ lines rule out a role as Notch signaling, given that glp-1(gf) results in a tumorous germ line? I understand the evidence in favor of the FBF pathway, but not why that observation specifically rules out Notch signaling. For example, the phenotype makes sense if GLP-1 was negatively regulating those genes…. (I’m not disagreeing with the statement, just feel like I’m missing something there)
On my review copy the table in figure 3 was very difficult to read.
Author Response
This is a well-written and incisive review of the C. elegans germline. The description of the somatic gonad is well done, pointing out recent work on the sheath cells that changes some of how we think about the somatic cells associated with the gonads. The review synthesizes a lot of complicated genetic data and makes sense of it. I have only a few minor comments that are intended to make the work accessible to a wider audience.
Thank you for these kind words; I found the following suggestions helpful and incorporated them all.
In Figure 1 panel C is not particularly informative, I understand the idea, but at first I thought it might be the male, then realized that it was only one of the gonads. Given the rich information about the structure and the exceptional image of the DTC/Sheath, panel C stands out. I would argue that the text above does a sufficient job of describing the migration, and that panel C is unnecessary. Alternatively, simply having arrows inside the worm that describe the path might be easier to interpret than what seems to be a pictorial time-lapse
Thank you very much, I agree and have removed this figure panel.
I think an additional image describing the domains of the germline might be helpful, to be able to visualize the progenitor, transition and meiotic regions.
This is a helpful suggestion, and to solve this and the last problem mentioned by this reviewer, I have moved the illustration of Figure 3 to the end of Figure 1.
Please define GLP NICD (Line 165) when it first appears.
I apologize for this omission and have now introduced the full phrase “Notch intracellular domain (NICD)” in this line.
In lines 216-217, why does the quadruple mutant forming tumorous germ lines rule out a role as Notch signaling, given that glp-1(gf) results in a tumorous germ line? I understand the evidence in favor of the FBF pathway, but not why that observation specifically rules out Notch signaling. For example, the phenotype makes sense if GLP-1 was negatively regulating those genes…. (I’m not disagreeing with the statement, just feel like I’m missing something there)
You are correct; an omission was introduced during editing. It was not the tumor but the continued expression of a Notch transcriptional reporter in this genetic background that implicates SYGL-1 and LST-1 as downstream targets that are not required for Notch pathway function. This has been simplified to state: “These genes do not encode components of Notch signal transduction [52].”
On my review copy the table in figure 3 was very difficult to read.
Figure 3 will now just be a table display item with the visual moved to Figure 1.
Reviewer 2 Report
This comprehensive review gives a detailed overview of our current understanding of the molecular and cellular mechanisms that drive germ line development in C. elegans. As the author notes, this is an important model system for the study of the biology of stem-cell compartments; arguably the only one present in this nematode. The review is thus an important contribution to the field of stem-cell biology in general, as well as being relevant for workers studying this model system. While there are several good overviews of C. elegans germ line development, the author’s account synthesises the important elements of the topic into a single source, and importantly highlights the remaining challenges and open problems for future studies.
Detailed experimental scrutiny of this system has not surprisingly revealed considerable complexity in terms of the underlying molecular signalling (including taxonomically-restricted components) and distinct spatiotemporal cellular states. The review gives an excellent account of these and places them in the context of our evolving understanding of the processes involved in germ line maturation and development.
I have only two suggestions to modify the current manuscript, but neither should preclude publication in its current form. Firstly, it would be nice to expand the evolutionary perspective of the review. For instance, how does germ line and stem-cell biology in C. elegans compare to other nematodes? There are a limited number of such studies, which means that this material could be relatively easily incorporated into a modified manuscript. In addition, comparing the molecular pathways to non-nematode animal models would also highlight potential conserved mechanisms and add value for readers studying other systems.
My second suggestion would be to replace the rather generic, technique-focussed section with one that summarises the key areas for future study. The content of this section in the current manuscript is the weakest part and seems like an afterthought; the justifications of the techniques could be applicable to almost any area of C. elegans research.
Author Response
This comprehensive review gives a detailed overview of our current understanding of the molecular and cellular mechanisms that drive germ line development in C. elegans. As the author notes, this is an important model system for the study of the biology of stem-cell compartments; arguably the only one present in this nematode. The review is thus an important contribution to the field of stem-cell biology in general, as well as being relevant for workers studying this model system. While there are several good overviews of C. elegans germ line development, the author’s account synthesises the important elements of the topic into a single source, and importantly highlights the remaining challenges and open problems for future studies.
Detailed experimental scrutiny of this system has not surprisingly revealed considerable complexity in terms of the underlying molecular signalling (including taxonomically-restricted components) and distinct spatiotemporal cellular states. The review gives an excellent account of these and places them in the context of our evolving understanding of the processes involved in germ line maturation and development.
Thank you, I’m glad you found this to be a useful synthesis of a subject about which much has been written. I wish I could extend it in the directions you suggest, below, but I feel at over 8000 words I am already testing limits.
I have only two suggestions to modify the current manuscript, but neither should preclude publication in its current form. Firstly, it would be nice to expand the evolutionary perspective of the review. For instance, how does germ line and stem-cell biology in C. elegans compare to other nematodes? There are a limited number of such studies, which means that this material could be relatively easily incorporated into a modified manuscript. In addition, comparing the molecular pathways to non-nematode animal models would also highlight potential conserved mechanisms and add value for readers studying other systems.
I share your interest in the evolution of this system. I have now included a reference to a striking putative conservation in the distal position of the gonadal sheath—the filarial parasite Brugia malayi also has sheath cells that abut the DTC (Foray et al., 2018, and Landmann et al., 2012). Given our current circumstances, however, I just am not able to do justice to a new section of the literature review unfortunately. As for the non-nematode animals, I refer readers to a recent expansive review of the C. elegans germline by Hubbard and Schedl, 2019. Because the issue focuses on C. elegans, and in the interest of time and space, I feel compelled not to delve too deeply into comparative arguments.
My second suggestion would be to replace the rather generic, technique-focussed section with one that summarises the key areas for future study. The content of this section in the current manuscript is the weakest part and seems like an afterthought; the justifications of the techniques could be applicable to almost any area of C. elegans research.
This is a very helpful suggestion, and I have removed this as a stand-alone section and folded discussion of the cited advances into other sections.
Reviewer 3 Report
Summary
This review article is a modern update on recent advances to understand germ stem cell biology in C. elegans. The past 10 years have seen huge advances in this field. Crispr/Cas9 gene editing has permitted the generation of fluorescent, endogenous tagged proteins. New and sophisticated live-imaging methodologies have allowed increasing insights into in vivo cellular and developmental events. This review summarizes these genetic, cell biological, and anatomical data, and proposes a novel model to account for the proliferation/differentiation choice made by germ stem cells.
Broad comments:
1. The most notable strength of this review is that it allows discussion of an important recent finding by the author, that germ line stem cells (GSCs) of C. elegans exist in distinct populations defined by their contact with two somatic cells: the distal tip cell (DTC) and the distal gonad sheath cell (Sh1). Furthermore, GSC proliferation and exit from the stem cell niche appears to be regulated by contact with these somatic cells. These findings are highly novel, and while yet unpublished [1], the discussion of their implications makes this review significant and impactful to the research community.
The novelty of the unpublished findings reviewed here also presents a challenge for the author. Overall, I felt this review could emphasize this innovation more! Specifically, it needed a more explicit acknowledgement that the author is presenting both new information and proposing a new model to account for GSC population behavior.
Below are specific comments that would address this criticism.
a. In section 2.1.1. Hermaphrodite Gonad Structure, lines 64-75 describe somatic sheath cell structure. Given that the structure of sheath cell Sh1 is new information, this discussion warrants a separate section, similar to that written for distal tip cells (section 2.1.3). Within the description of Sh1 structure (lines 64-75), I recommend a statement to acknowledge that this new information leads to a new model, to be described later in this review.
b. In 2.3.2, the author describes a new model. However, this section is in need of a more complete description of the new findings that form the basis of that model. I say this because I had to read the attached unpublished work in order to understand what was meant by Lines 381-382 “The discovery of asymmetric germ cell divisions across the DTC-Sh1 boundary...” This is critical. A detailed description of the anatomical distinctions would certainly help the reader when, in the second paragraph of 2.3.2, the author describes how the populations may also be genetically distinct.
References cited:
[1] Gordon, K.L.; Zussman, J.W.; Li, X.; Martin, C.M.; Sherwood, D.R. Stem cell niche exit in C. elegans via orientation and segregation of daughter cells by a cryptic cell outside the niche. eLife, Revis. 2020.
Minor, specific comments on writing:
Line 68: The term and acronym “gonadal BM” was not previously defined.
Line 71: The function/ortholog of gene cacn-1 is not explained, so it is not clear what the significance of this statement is for sheath cell biology.
Line 137: The author writes “glp-1(gf)” without an explanation that glp-1 is a C. elegans Notch, or what is meant by “gf.”
Line 149: What is meant by “mitotic gene?” In other cell contexts, I would not consider Notch/GLP-1 to be a “mitotic gene.”
Lines 216-217: The author does not explain why the quadruple mutant demonstrates that genes lst-1 and sygl-1 are not required for Notch signal transduction. Surely this conclusion requires a comparison to another genotype. (The gld-1; gld-2 mutant?) But this is not described.
Line 221: A missing word (underlined) at “one of the target genes”
Line 229: The word “solved.” Truly?
Line 280: An extra word (underlined) at “lose are no longer marked by”
Figure 3A: These labels on the drawing would be very helpful: Distal-most GSCs, Interface GSCs, Proximal PZ cells.
Figure 3B: The font is extremely small in size, difficult to read.
Line 414: The reader is allowed to infer for themselves which of the germ cell populations is similar to transit amplifying stem cells. (Which is it?) The author should specify at the beginning of this paragraph. What are the defining features of a transit amplifying population? This information would help the reader interpret the evidence reviewed throughout this section.
Author Response
This review article is a modern update on recent advances to understand germ stem cell biology in C. elegans. The past 10 years have seen huge advances in this field. Crispr/Cas9 gene editing has permitted the generation of fluorescent, endogenous tagged proteins. New and sophisticated live-imaging methodologies have allowed increasing insights into in vivo cellular and developmental events. This review summarizes these genetic, cell biological, and anatomical data, and proposes a novel model to account for the proliferation/differentiation choice made by germ stem cells.
Broad comments:
- The most notable strength of this review is that it allows discussion of an important recent finding by the author, that germ line stem cells (GSCs) of C. elegans exist in distinct populations defined by their contact with two somatic cells: the distal tip cell (DTC) and the distal gonad sheath cell (Sh1). Furthermore, GSC proliferation and exit from the stem cell niche appears to be regulated by contact with these somatic cells. These findings are highly novel, and while yet unpublished [1], the discussion of their implications makes this review significant and impactful to the research community.
The novelty of the unpublished findings reviewed here also presents a challenge for the author. Overall, I felt this review could emphasize this innovation more! Specifically, it needed a more explicit acknowledgement that the author is presenting both new information and proposing a new model to account for GSC population behavior.
Thank you for your enthusiasm about these recent findings! I have been given the advice to be moderate in my statements of novelty. I hope the changes I have made in the revised manuscript and the move of the illustration from Figure 3 to Figure 1 make the change in the model more explicit without seeming overexuberant.
Below are specific comments that would address this criticism.
- In section 2.1.1. Hermaphrodite Gonad Structure, lines 64-75 describe somatic sheath cell structure. Given that the structure of sheath cell Sh1 is new information, this discussion warrants a separate section, similar to that written for distal tip cells (section 2.1.3). Within the description of Sh1 structure (lines 64-75), I recommend a statement to acknowledge that this new information leads to a new model, to be described later in this review.
This is very helpful advice for structuring the argument, thank you. The end of this paragraph now reads: “Below, this review will further explore how the discovery of Sh1 making extensive contacts with the PZ calls for the formulation of a new model for germ stem cell compartmentalization.”
- In 2.3.2, the author describes a new model. However, this section is in need of a more complete description of the new findings that form the basis of that model. I say this because I had to read the attached unpublished work in order to understand what was meant by Lines 381-382 “The discovery of asymmetric germ cell divisions across the DTC-Sh1 boundary...” This is critical. A detailed description of the anatomical distinctions would certainly help the reader when, in the second paragraph of 2.3.2, the author describes how the populations may also be genetically distinct.
Thank you for this suggestion; a depiction of a dividing interface germ cell has been added to Figure 1 along with the rest of the model from Figure 3.
References cited:
[1] Gordon, K.L.; Zussman, J.W.; Li, X.; Martin, C.M.; Sherwood, D.R. Stem cell niche exit in C. elegans via orientation and segregation of daughter cells by a cryptic cell outside the niche. eLife, Revis. 2020.
Minor, specific comments on writing:
Line 68: The term and acronym “gonadal BM” was not previously defined.
Thank you, this term has now been expanded to “gonadal basement membrane”
Line 71: The function/ortholog of gene cacn-1 is not explained, so it is not clear what the significance of this statement is for sheath cell biology.
This reference has been expanded to read: “Sheath cells require CACN-1/Cactin for proper splicing and development, without which worms are sterile [17].”
Line 137: The author writes “glp-1(gf)” without an explanation that glp-1 is a C. elegans Notch, or what is meant by “gf.”
Thank you for pointing this out; the jargon has been corrected. “Conversely, constitutive activation of the Notch receptor in a glp-1(gf) gain-of-function mutant causes a tumorous germ line full of mitotically dividing cells that never enter meiosis [43,44].”
Line 149: What is meant by “mitotic gene?” In other cell contexts, I would not consider Notch/GLP-1 to be a “mitotic gene.”
It is a convention to refer to the undifferentiated germ cells as “mitotic” because they have not entered meiosis; I have tried to avoid this terminology in the review and appreciate your note. I have rephrased to refer to “genes that promote the undifferentiated cell fate” in this section.
Lines 216-217: The author does not explain why the quadruple mutant demonstrates that genes lst-1 and sygl-1 are not required for Notch signal transduction. Surely this conclusion requires a comparison to another genotype. (The gld-1; gld-2 mutant?) But this is not described.
You are correct; an omission was introduced during editing. It was not the tumor but the continued expression of a Notch transcriptional reporter in this genetic background that implicates SYGL-1 and LST-1 as downstream targets that are not required for Notch pathway function. This has been simplified to state: “These genes do not encode components of Notch signal transduction [52].”
Line 221: A missing word (underlined) at “one of the target genes”
Thank you, now reads: “One of the target mRNAs”
Line 229: The word “solved.” Truly?
Fair enough! It now reads: “The direct Notch signaling targets that interact with FBF to prevent meiotic entry have been identified; other questions remain.”
Line 280: An extra word (underlined) at “lose are no longer marked by”
Thank you, this typo has been corrected: “…cells proximal to this interface are no longer marked by…”
Figure 3A: These labels on the drawing would be very helpful: Distal-most GSCs, Interface GSCs, Proximal PZ cells.
These labels have been added, and this figure panel has been moved to Figure 1E.
Figure 3B: The font is extremely small in size, difficult to read.
Apologies, this will now appear as Table 1 which will hopefully improve legibility.
Line 414: The reader is allowed to infer for themselves which of the germ cell populations is similar to transit amplifying stem cells. (Which is it?) The author should specify at the beginning of this paragraph. What are the defining features of a transit amplifying population? This information would help the reader interpret the evidence reviewed throughout this section.
Thank you for this suggestion, the discussion has been expanded to clearly state that the non-stem progenitor cells are analogous to a TA population.